# Pathological and genetic aspects of spontaneous mammary gland tumor in *Tupaia belangeri* (tree shrew)

Chi Hai-Ying[1☯], Yuki Tanaka[2☯], Tatsuro Hifumi[2☯], Koichiro Shoji[3], Mohammad Enamul Hoque Kayesh[1,4], Md Abul Hashem[1,5], Bouchra Kitab[1], Takahiro Sanada[6], Tomoko Fujiyuki[3], Misako Yoneda[3], Hitoshi Hatai[2], Akira Yabuki[7], Noriaki Miyoshi[2], Chieko Kai[3], Michinori Kohara[6]*, Kyoko Tsukiyama-Kohara[1]*

1 Joint Faculty of Veterinary Medicine, Transboundary Animal Diseases Centre, Kagoshima University, Kagoshima, Japan, 2 Department of Pathology, Joint Faculty of Veterinary Medicine, Kagoshima University, Kagoshima, Japan, 3 Laboratory of Experimental Animals, The Institute of Medical Science, The University of Tokyo, Tokyo, Japan, 4 Department of Microbiology and Public Health, Faculty of Animal Science and Veterinary Medicine, Patuakhali Science and Technology University, Barishal, Bangladesh, 5 Department of Health, Chattogram City Corporation, Chattogram, Bangladesh, 6 Department of Microbiology and Cell Biology, Tokyo Metropolitan Institute of Medical Science, Tokyo, Japan, 7 Department of Clinical Pathology, Joint Faculty of Veterinary Medicine, Kagoshima University, Kagoshima, Japan

☯ These authors contributed equally to this work.
* kohara-mc@igakuken.or.jp (MK); kkohara@vet.kagoshima-u.ac.jp (KTK)

**Data Availability Statement:** All data necessary to replicate our study's findings is available in the paper and its supporting files.

## Abstract

Mammary gland cancer is the most common cancer occurring in women globally. Incidences of this cancer in Japan are on the increase. Annually, more than 70,000 new cases are recorded in Japan and about 1.7 million in the world. Many cases are still difficult to cure completely, and animal models are required for the characterization of the biology, therapeutic strategy, and preventive measures for spontaneous mammary tumor. The mouse model used currently has some limitations owing to structural differences between mouse and human mammary glands. *Tupaia belangeri* (tree shrew), which belongs to the Tupaiidae family, shows relatively high genetic homology and structural similarity to human mammary glands. Here, we characterized the spontaneous mammary tumors in 61 female tree shrews of different ages. The incidence rate was 24.6% (15/61), and the rate of simultaneous or metachronous multiplex tumors was 60% (9/15). From the incidence pattern, some cases seemed to be of familial mammary gland tumor, as the offspring of female tree shrews No. 3 and 9 and male tree shrew No. 11 showed a high incidence rate, of 73.3% (11/15). Average incidence age for tumor development was 2 years and 3 months, and the earliest was 10 months. Histochemical analysis indicated that spontaneous mammary gland tumors in the tree shrew show the features of intraductal papillary adenomas (22 cases), except 2 tubulopapillary carcinoma cases (No. 75 and 131). All the cases were positive for the progesterone receptor, whereas 91.3% were positive for the estrogen receptor, and 4.3% were HER-2 positive. We have also confirmed the expression of nectin-4 in some mammary tumor cells. Additionally, we subjected tree shrews to cytodiagnosis or X-ray CT. Thus, the findings of this study highlight the potential of the tree shrew as a valuable new animal model for mammary gland tumor study.

**Funding:** the Japan Agency for Medical Research and Development, the Tokyo Metropolitan Government; the Ministry of Health and Welfare of Japan; and the Ministry of Education, Culture, Sports, Science and Technology of Japan.

**Competing interests:** The authors have declared that no competing interests exist.

## Introduction

Mammary gland cancer is among the most prevalent forms of cancer in women [1]. In Japan, more than 70,000 women are diagnosed of mammary gland cancer every year [2].

However, not all the patients can be cured at present. The development of suitable animal models for understanding the underlying mechanism as well as for identifying novel preventive and therapeutic approaches are required.

At present, rodents are used in most of the studies associated with breast cancers [3], although they are different from human beings in terms of both the biological characteristics and molecular mechanisms underlying development and progression of breast cancers, which limit their application in breast cancer research.

*Tupaia belangeri* is a small non-primate mammal belonging to the family Tupaiidae, which has a body weight of approximately 150 g [4]. In comparison with rodents, the genome sequence of tree shrew shows a higher homology with that of primates [4–6]. The tree shrew shows susceptibility to hepatitis viruses infections (hepatitis C virus [7, 8], hepatitis B virus [9, 10]) and can be used for the evaluation of anti-depressant drugs [11]. In addition, the development and physiological characters of the tree shrew's mammary gland are similar to those of human beings [12, 13].

A previous study observed incidences of tumors (intraductal mammary adenocarcinoma) in the tree shrew's mammary glands (5/1,132, 0.44%) [14]. To characterize spontaneous mammary tumor in a tree shrew model, we investigated the morphogenesis of spontaneous mammary tumor in the tree shrew by pathologic analysis, and characterized its lineage.

## Materials and methods

### Animals

The tree shrews used in this study were originally obtained from the Laboratory Animal Center at the Kunming Institute of Zoology and Chinese Academy of Sciences, Kunming, China (10 male and 20 female) and breed in our animal facility. A total of 61 female tree shrews were used in this study. This study was carried out following the Guidelines of Animal Experimentation of the Japanese Association for Laboratory Animal Science and Guide for the Care and Use of Laboratory Animals of the National Institutes of Health. All experimental protocols were approved by the institutional review boards of the regional ethics committees of Kagoshima University (VM15051 and VM13044).

The animals were individually housed in cages and fed a daily regimen of eggs, fruit, water, and dry marmoset food (CMS1M, Crea). The animals were humanely handled in accordance with the Institutional Animal Care and Use Committee for Laboratory Animals. During surgery, animals were anesthesized by intramuscular injection of ketamine (50 mg/kg) and atropine (0.33 mg/kg) and anesthesia was maintained with 2% isoflurane (Fujifilm-Wako Co.). Mammary tumors were detected by eye observation every day, and mammary tumors with more than 1 cm in diameter were subjected to surgical excision. After surgery, animals were returned to Laboratory Animal Center.

### Cytodiagnosis, histochemical analysis, and immunohistochemistry staining

Mammary tumor in tree shrews was diagnosed by cytodiagnosis and histology after surgery. Excised tissues were stained with May-Giemsa (Sigma-Aldrich Co.) for cytodiagnosis. Also, the tissues were fixed with 10% neutral buffered formalin solution (Wako), and stained with hematoxylin and eosin (H&E) for tissue histology. The Histofine Simple Stain MAX-PO Multi system (Nichirei) was used for immunohistochemical staining. The following primary

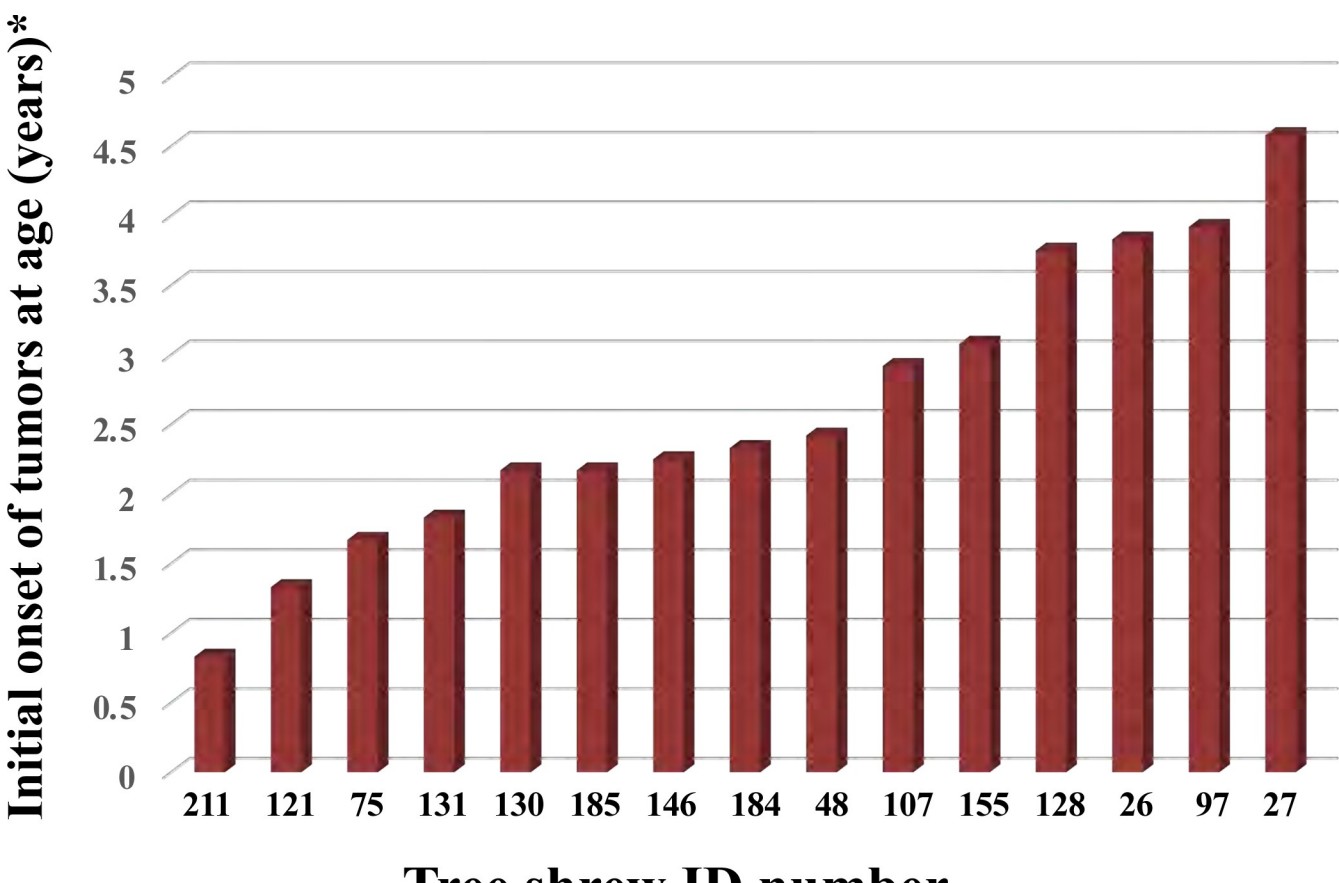

**Fig 1. Onset of mammary tumors in female tree shrews was indicated.** Initial onset age of spontaneous mammary tumors in tupaia is indicated. Each tupaia ID number is indicated on the horizontal line.

antibodies were used for immunohistochemistry staining: anti-estrogen receptor (ER)-α (1:50 dilution, mouse monoclonal antibody (MoAb) clone 1D5, DAKO), progesterone receptor (PR) (1:1,000, rabbit polyclonal, clone C-19, Santa Cruz), human epidermal growth factor receptor (HER)2 (1:250, rabbit MoAb clone A0485, DAKO), and Ki67 (1:50, mouse MoAb, clone MIB-1, DAKO) antibodies. Detected antigens were visualized with 3, 3′-diaminobenzine tetrahydrochloride reagent. Sections were counter stained with Mayer's hematoxylin. Histopathological diagnosis was made based on classification of mammary tumors in domestic animals [15]. Evaluation of ER, PR, HER-2, and Ki-67 positive ratio was performed as described in S1 File and S1 Table.

## Gene mutation analysis

Gene mutations in tumor tissues were analyzed as described previously [12]. Briefly, total RNA from tumor tissues was extracted using an ISOGEN Kit (Nippon Gene) according to the manufacturer's instructions. cDNA was synthesized using a SuperScript III Reverse Transcriptase kit (Invitrogen) following the manufacturer's instructions. Gene-specific PCR targeting of PTEN and PIK3CA genes was performed, and the amplified product was subcloned and sequenced. The primers used to amplify the candidate genes PTEN and PIK3CA were as follows: PTEN forward 5′-ATGACAGCCATCATCAAAGAG-3′ and reverse 5′-TCAGACTTTTG

**Table 1. Summary of tree shrews with mammary tumors.**

| Tree shrew No. | Date of surgery | Date of birth | Age at surgery | Histology | Markers[3] |
|---|---|---|---|---|---|
| No.26 | 2016/1/19 | 2012/2/26 | 3Y 10 M[4] | IPA[1] | ER, PR, Ki67 |
| | 2016/2/4 | | 3Y 11 M | IPA | ER, PR, Ki67, Her2 |
| | 2016/8/30 | | 4Y 6 M | IPA | ER, PR, Ki67 |
| | 2017/1/26 | | 4Y 11M | IPA | ER, PR, Ki67, Her2 |
| No.27 | 2017/1/6 | 2012/4/11 | 4Y7 M | NT* | NT |
| | 2017/2/9 | | 4Y 8 M | | |
| | 2017/9/12 | | 5Y 5 M | | |
| No.48 | 2016/1/19 | 2013/8/5 | 2Y 5M | IPA | ER, PR, Ki67 |
| No.75 | 2015/6/16 | 2013/9/20 | 1 Y 8 M | NT | NT |
| | 2016/6/19 | | 4Y 3M | IPA | ER, PR, Ki67 |
| | 2018/2/20 | | 6Y 1M[5] | TC[2] | ER, PR, Ki67, Her2 |
| No.97 | 2017/10/26 | 2013/11/9 | 3Y 11M | IPA | ER, PR, Ki67 |
| | 2017/10/31 | | 3Y 11 M | IPA | PR, Ki67 |
| | 2018/01/15 | | 4Y 2 M | IPA | ER, PR, Ki67 |
| | 2018/01/30 | | 4Y 2 M | IPA | PR, Ki67 |
| No.107 | 2017/1/10 | 2014/2/27 | 2 Y 11 M | NT | NT |
| No.121 | 2015/10/14 | 2014/5/30 | 1Y4 M | IPA | ER, PR, Ki67 |
| | 2017/8/1 | | 3Y 2 M | IPA | PR, Ki67 |
| | 2017/12/15 | | 3Y 9M | IPA | ER, PR, Ki67 |
| No.128 | 2018/01/15 | 2014/06/20 | 3Y 9 M | IPA | ER, PR, Ki67 |
| No.130 | 2016/8/30 | 2014/6/23 | 2Y 2 M | IPA | ER, PR, Ki67 |
| | 2016/9/21 | | 2 Y 3 M | NT | NT |
| No.131 | 2016/4/13 | 2014/6/23 | 1Y 10 M | NT | NT |
| | 2017/6/2 | | 3 Y | IPA | PR, Ki67 |
| | 2018/2/14 | | 3Y 7M | IPA | ER, PR, Ki67, Her2 |
| | 2019/12/13 | | 5Y 6M | TC | NT |
| No.146 | 2017/1/6 | 2014/10/24 | 2 Y 3 M | NT | NT |
| No.155 | 2017/12/15 | 2014/10/30 | 3Y 1 M | IPA | ER, PR, Ki67 |
| No.184 | 2017/6/14 | 2015/2/17 | 2Y 4M | IPA | ER, PR, Ki67 |
| | 2018/2/20 | | 3Y | IPA | ER, PR, Ki67 |
| No.185 | 2017/4/19 | 2015/2/17 | 2 Y 2M | NT | NT |
| | 2017/10/19 | | 2 Y 8M | IPA | PR, Ki67 |
| No.211 | 2016/4/27 | 2015/6/25 | 10 M | IPA | ER, PR, Ki67 |

[1] Intraductal papillary adenoma

[2] Tubulopapillary carcinomas

[3] Markers with positive reaction

[4] Year; Y, Months; M

[5] found dead and indicated with red colored text

*NT: not tested

TAATTTGTG-3′; PIK3CA forward 5′-ATGCCTCCACGACCAT CA-3′ and reverse 5′-TCA GTTCAATGCATGCTG-3′.

## Flow cytometric analysis

Tumor tissues were digested with Hank's balanced salt solution (HBSS; Life Technologies) containing 5 mM 4-(2-hydroxyethyl)-1-piperazine ethanesulfonic acid (HEPES), 2% fetal

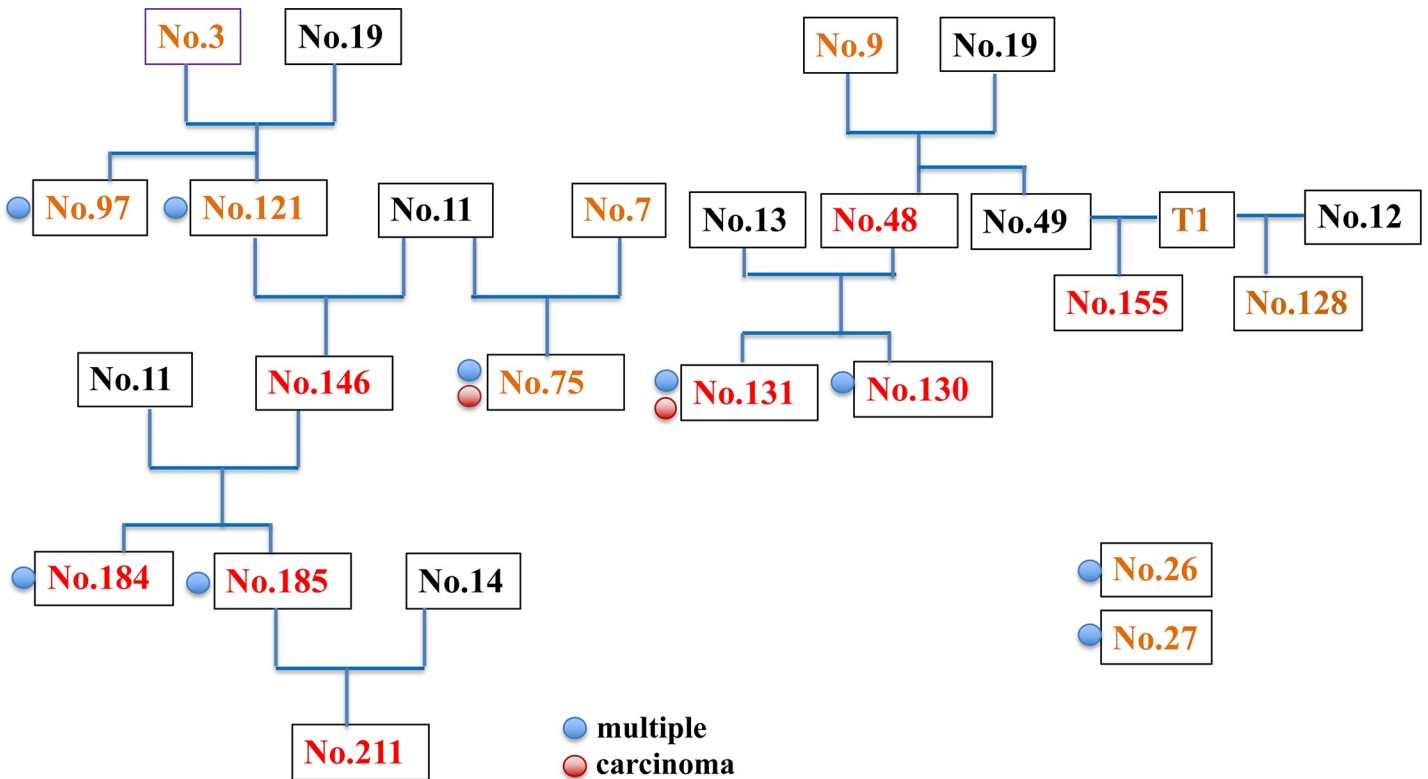

**Fig 2. Lineage of tree shrew parents and offsprings.** Brown color indicates female and black color indicates male tree shrew. Tree shrew numbers with red color indicate the possible familial mammary tumor and blue circle indicates the simultaneous or metachronous multiple tumors. Red circle indicates the incidence of malignant carcinoma.

bovine serum (FBS), 1 mg/mL collagenase (Wako Pure Chemical Industries), and 0.1% DNase I (Life Technologies), as described [16]. A total number of $10^6$ cells were labeled with 0.5 μg of primary Ab (anti-nectin-4 goat polyclonal Antibody (R&D systems)) or isotype control in 100 μL of HBSS (Life Technologies) containing 2% FBS and HEPES, and then with an Alexa-Fluor-488-conjugated anti-goat IgG Ab (Life Technologies) diluted 1:500. To exclude dead cells, 7-AAD (Beckman Coulter) was used in the analysis. The cells were analyzed with a BD FACS Verse flow cytometer (BD Co.), and the data were processed using the Flow Jo FACS analysis software ver. 9.5.3.

## X-ray CT analysis

X-ray CT analysis of tree shrews was performed using Latheta LCT-200 (Hitachi Co. Ltd.) under anesthesia.

## Results

### Onset of mammary gland tumors

We characterized 61 female tree shrews, out of which 15 were found to have mammary tumors (Fig 1). The surgical excision history of mammary gland tumors is summarized in Table 1. The incidence rate of mammary tumors was 24.6% (15/61) and the average age of tumor incidence was 2 years 3 months. The simultaneous or metachronous case of mammary tumors was observed in 9 tree shrews (No.121, 75, 97, 130, 131, 184, 185, 26, and 27) and the rate was

60.0% (9/15). The mammary tumor usually occurs in two- to three-year old virgin tree shrews. The occurrence of mammary tumor in a 10-month-old tree shrew (No. 211) indicates its existence in juvenile individuals.

### Influence of inheritance on mammary gland tumors in tree shrews

The parent-child relationship in the tree shrew was summarized in Fig 2. Mammary tumor was detected in 66.7% (4/6) of the offspring of No. 3. However, all (100%) of the offspring of No. 9 (5/5, No. 48, 128, 130,131, 155) and 11 (5/5, No. 75, 146, 184, 185, 211) developed mammary tumors. These high incidence rate of mammary gland tumors is indicative of the character of familial cancers [17] and possible influence of inheritance factors. Tree shrews No. 3 and 9 did not develop mammary tumors; therefore, unknown factor(s) inherited from the male may have influenced the incidences in the offspring.

**A**

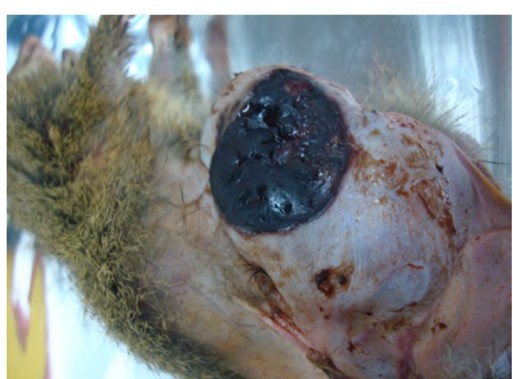

**C**

**B**

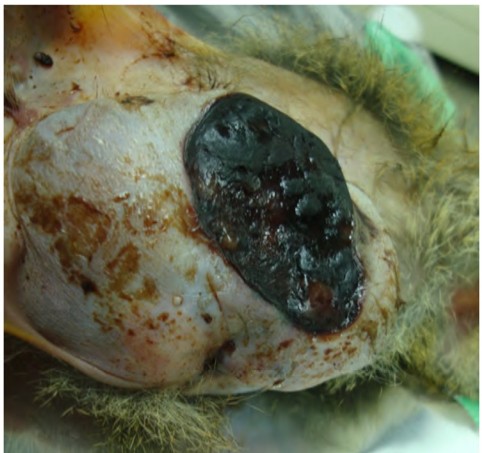

**Fig 3.** (**A**) Outlook of tree shrew spontaneous mammary tumor (No.75 intraductal papillary adenomas, 4 yeas 3 months old), (**B**) tubulopapillary carcinoma (No.75, 6 year 1 month old), and (**C**) cytodiagnosis of tree shrew No.26 mammary gland tumor (3 year 10monts old). Tumor cells are showing features of malignant epithelial tumor.

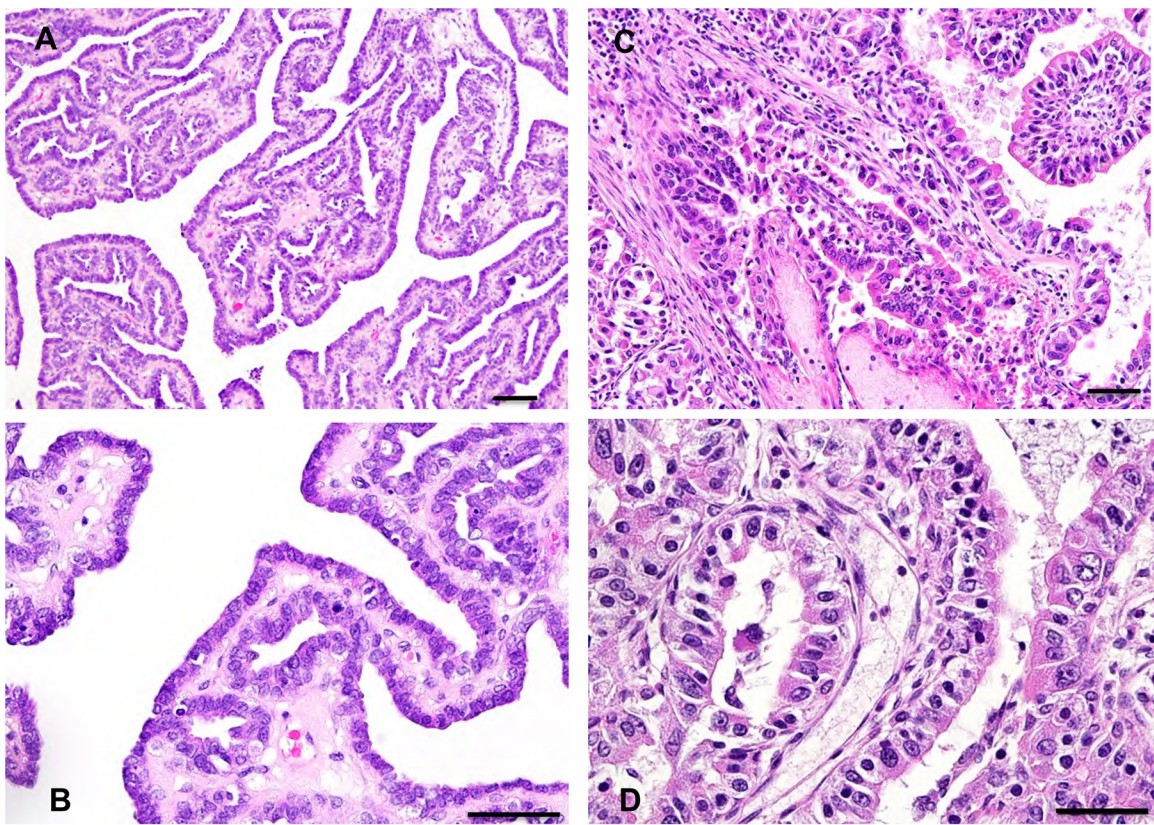

**Fig 4. HE staining of tree shrew mammary gland tumors (bar = 100μm).** Intraductal papillary adenomas (tree shrew No.97, 3 years 11 months old, **A, B**) and tubulopapillary carcinoma in tree shrew No.75 (6 years 1 month old, **C, D**).

### Characteristics of spontaneous mammary gland tumors in tree shrews

The mammary tumors in tree shrews were oval or irregular, with smooth surface, and were frequently complicated with ulcer (No. 75: 1 year and 8 months, Fig 3A). A malignant case with large tumors with ulcer was recorded (No. 75: 6 years and 1 month, Fig 3B). Cytodiagnosis suggested tree shrew mammary tumors to be malignant epithelial tumors (No. 26, Fig 3C). Mammary tumor with sheet-like clump and strong conjunction of tumor cells suggested the epidermal origin of cell clump. Mammary tumor cells were basophilic and their nucleus showed atypical morphology (in terms of size and nucleolus size).

Based on histopathological examinations, mammary gland tumors were divided into following 2 types (Fig 4): intraductal papillary adenomas were observed in 22 cases (Fig 4A and 4B, Table 2) and tubulopapillary carcinomas were observed in 2 cases (No. 75, 131; Fig 4C and 4D, Table 2). Intraductal papillary adenomas had an intraductal, papillary, arborescent growth pattern in which the neoplastic epithelial cells were supported by a fibrovascular stalk. The neoplastic epithelial cells of the papillae, which form a single to double layer, have an oval nucleus and a small amount of cytoplasm. A layer of inconspicuous flattened myoepithelial cells was observed below the neoplastic epithelial cells. Anisokaryosis and anisocytosis were minimal, with low mitotic activity. Tubulopapillary carcinomas were composed of neoplastic epithelial cells arranged in a tubular and papillary pattern. Nuclear/cytoplasm ratio of tumor cells was high, and nucleus showed size variation and atypical morphology. The tumor cells had high mitotic activities and invaded the surrounding mammary tissues (Fig 4C and 4D) [18]. Pulmonary metastasis was observed in one case (No. 131, at 5 year 6 months of age).

**Table 2. Characteristics of tree shrew mammary gland tumors.**

| Tree shrew No. | Age at surgery | Histology | Position# | ER* | PR* | Ki67* | HER-2 |
|---|---|---|---|---|---|---|---|
| No.26 | 3Y10 M | IPA*1 | LR3 | 3a | 3b | 11.8 | 0 |
| | 3Y 11 M | IPA | R1 | 3a | 3b | 9.7 | 1+ |
| | 4Y 6 M | IPA | R3 | 3b | 3b | 27.2 | 1+ |
| | 4Y 11 M | IPA | MG | 3a | 3b | 18 | 0 |
| No.48 | 2Y 5 M | IPA | L3 | 3b | 3b | 33.4 | 0 |
| No.75 | 4 Y 3 M | IPA | PN | 3b | 3b | 15.3 | 0 |
| | 6 Y 1 M | TC | MG | 1 | 2 | 68.1 | 2+ |
| No.97 | 3 Y 11 M | IPA | L3 | 3a | 3b | 6.7 | 0 |
| | 3 Y 11 M | IPA | L1-2 | 1 | 3a | 6.1 | 0 |
| | 4 Y 2 M | IPA | L1 | 2 | 3b | 25.4 | 0 |
| | 4 Y 2 M | IPA | L2 | 1 | 3a | 22.6 | 0 |
| No.121 | 1 Y 4 M | IPA | L1 | 3a | 3b | 34.7 | 0 |
| | 3 Y 2 M | IPA | L1 | 0 | 3b | 7.4 | 0 |
| | 3 Y 9 M | IPA | LMG | 3a | 3b | 20.9 | 0 |
| No.128 | 3 Y 9 M | IPA | PN | 1 | 2 | 6.9 | 0 |
| No.130 | 2 Y 2 M | IPA | LR3 | 3b | 3b | 30.8 | 0 |
| No.131 | 3 Y | IPA | MG | 0 | 3b | 5.5 | 0 |
| | 3 Y 7 M | IPA | MG | 3b | 3b | 16.2 | 1+ |
| No.155 | 3 Y 1 M | IPA | R3 | 2 | 3b | 25.3 | 0 |
| No.184 | 2 Y 4 M | IPA | L2 | 1 | 3b | 7.9 | 0 |
| | 3yr | IPA | MG | 3b | 3b | 57.2 | 0 |
| No.185 | 2 Y 8 M | IPA | Last MG | 1 | 3b | 17.2 | 0 |
| No.211 | 10 M | IPA | 1–2 | 3b | 3b | 51.9 | 0 |

*1 Intraductal papillary adenomas

*2 Tubulopapillary carcinomas

#LR: left and right, MG: mammary gland, PN: pubic neighborhood, number: position of tumors

*J-score (S1 Table)

## Mammary gland tumor markers of tree shrews

Mammary tumor cells were reported to be stained with estrogen receptor α (ERα), progesterone receptor (PR) and HER-2 in human [19] as well as tree shrew [12, 13] (Table 2). Staining was done for these markers in tree shrew mammary tumor tissues (Fig 5). Out of 23 characterized cases, 21 were ER positive (91.3%) and 100% were positive for PR (Table 2, Fig 5). All cases were ER or PR positive; therefore, they could be classified as the luminal type [20]. Also in the malignant case (tree shrew No. 75), overexpression of HER-2 (Score 2+) was observed (Fig 5, Table 2). Labeling index (LI) measured based on Ki-67 positive tumor cells was 3.4–68.1% (Fig 5, Table 2).

Mutations in PTEN and PIK3CA genes [12] were not observed in mammary tumors from tupaias No.75, 97, 185, and 211, after sequencing.

In addition to these markers, Nectin-4 has been also reported to be expressed in breast cancer cells [21] and Nectin-4 targeted oncolytic measles virus (MV) vector has been constructed [22]. To clarify the possibility that MV vector targeting Nectin-4 is applicable for the tree shrew mammary tumor model, we stained mammary tumor cells with anti-Nectin-4 antibody (Fig 6). Thus, 27.4% of cells could be stained with anti-Nectin-4 antibody using FACS.

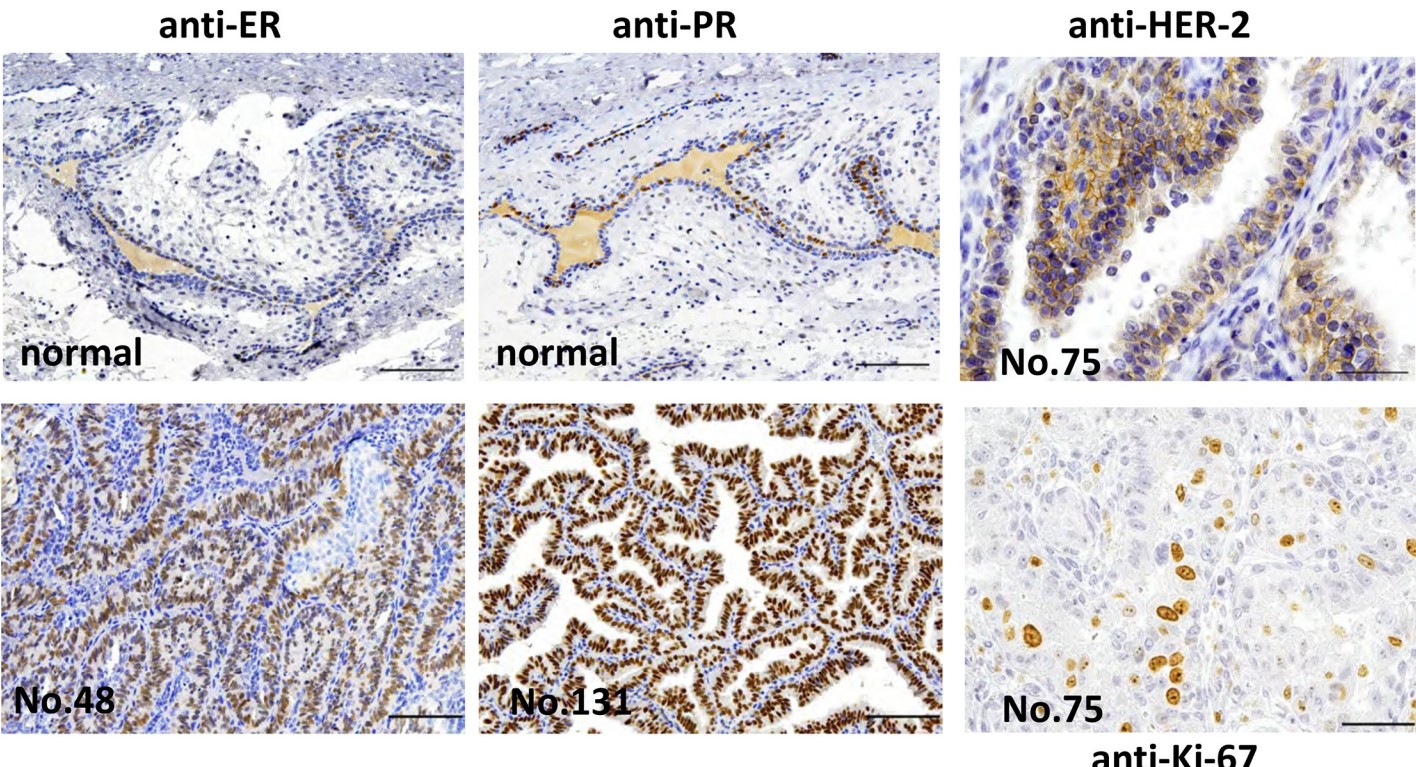

**Fig 5. Immunohistochemistry staining of tree shrew mammary gland tumors with anti ER (tree shrew No. 48, 2 years 5 months old) and anti-PR (tree shrew No. 131, 3 years 7 months old) antibodies (bar = 100μm) (lower left and middle), and tree shrew tubulopapillary carcinomas with anti-HER-2 (tree shrew No.75: 6 years and 1 month old) (bar = 50μm lower; ×400) (upper right).** More than 10% of the tumor cell membrane was stained partially or weakly but entirely (upper right). In normal tissues of the tree shrew mammary gland, the expression of ER and PR was observed in the nucleus of ductal epithelial cells (upper left and middle). Ki-67 + mammary tumor cells (tree shrew No.75, 6 years and 1 month old) are also shown (lower right; ×400).

### X-ray CT analysis

X-ray CT analysis was also performed on tree shrews with mammary tumor (No.26, Fig 7) using LCT-200 (Hitachi, Ltd., Tokyo, Japan) (S1A Fig). The entire bone and skull structures were also observed (S1B Fig, S1C Fig) and they did not show significant abnormality. Numerous sarcomas were observed in the outer layer of the skin (red arrows, Fig 7A). Intraperitoneally, there were endocrine tumors close to the liver and kidney (Fig 7B and 7C). Based on the position and pathologic features of the tumor in the intraperitoneal cavity, it was deduced as adrenocortical adenoma (S2 Fig).

### Discussion

The results in this study showed that tree shrews can develop spontaneous mammary tumors. The incidence rate of mammary tumors in female tree shrews was 24.6% (15/61). Findings based on the tree shrew lineage study suggest that inheritance should be a major determinant of the occurrence of spontaneous mammary tumors in tree shrews and the occurrence of juvenile breast carcinoma was also noted. The earliest onset of breast cancer in tree shrews was 10 months. A tree shrew of less than 4 years can be considered a juvenile, as tree shrews typically have a lifespan of about 7–8 years [4]. The factors responsible for the incidence of breast cancer in the juvenile are unclear at present, but inheritance might be a predisposing factor. Juvenile papillomatosis has been reported to be linked to strong family history, which may lead to the onset of proliferative diseases in women less than 30 years of age [23]. As malignant cases were

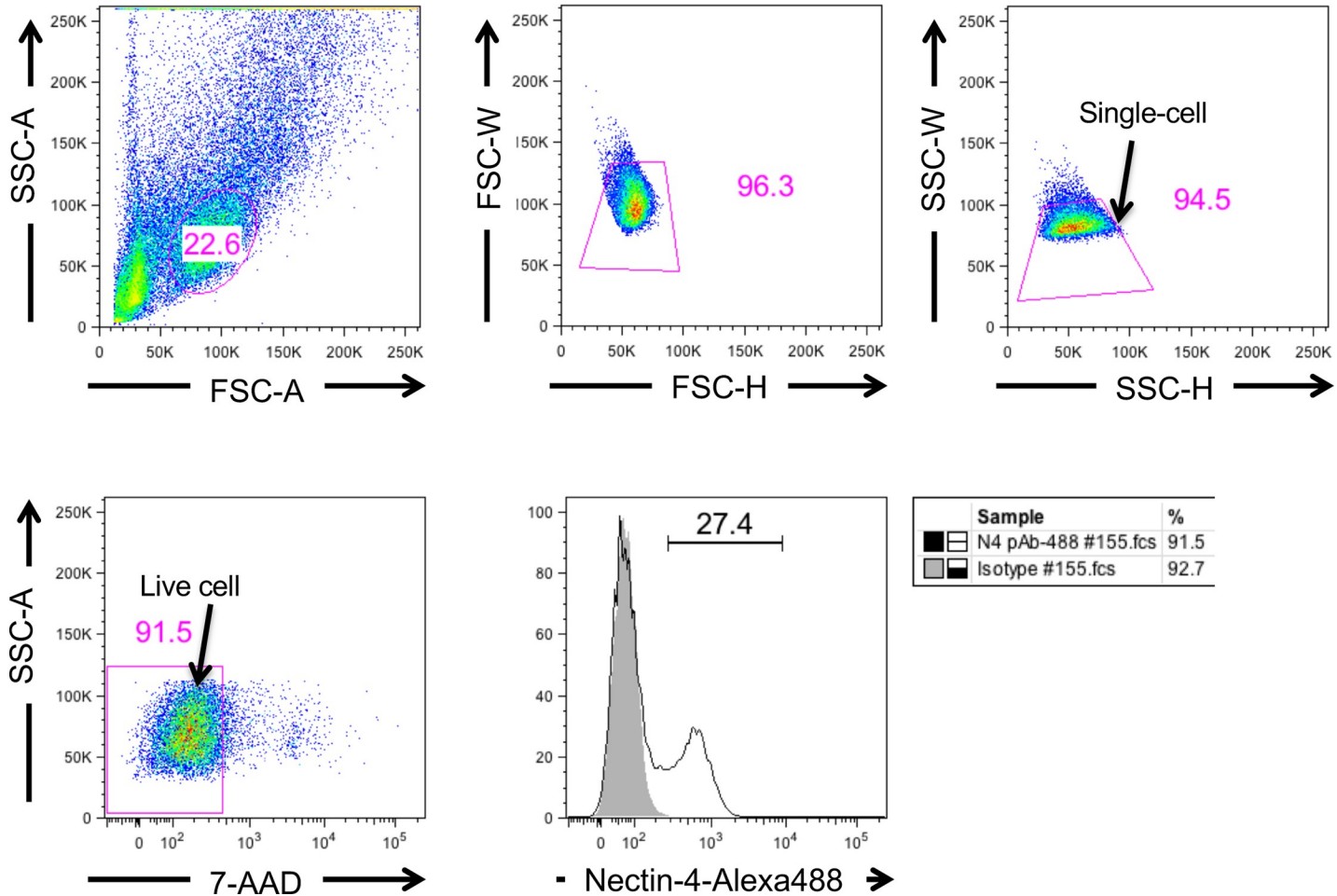

**Fig 6. Nectin-4 staining of breast cancer cells (tree shrew No.155, 3 years 1 month old).** Live single cells were calculated as 91.5 and 94.5% by size using the FACS verse. Nectin-4 positive cells were calculated to be 27.4% in living cells (lower right column).

only observed in older tree shrews (No. 131: 5 years and 6 months and No. 75: 6 years and 1 month), malignancy could be influenced by age dependent factor(s). A previous study reported the incidence rate of spontaneous mammary carcinoma in tree shrew as 0.44% in 4–6-year-olds [14]. Intraductal mammary carcinomas in tree shrews have also been reported [12, 24]. Therefore, the results in this study highlight the possibility that mammary gland tumors appearing in young tree shrews may turn malignant with age.

The positive ratio of ER was 63% and 100% tumor cases were PR positive, which was consistent with a previous report [12]. The variation in the ER and PR expression in the mammary gland tumors might be influenced by the periodical blood estrogen level that affects the expression of PR [25]. Also, a malignant case (No.75: 6 years and 1 month) showed HER-2 positive (Score 2+), clear invasion, atypical morphology of nucleus, and high growth activity (LI:68.1) indicating highly malignant mammary gland carcinoma. It has been reported that HER-2 overexpression is linked with malignancy [26]. In case No. 75, the expression of ER was negative and that of PR was low, showing that HER-2 expression might contribute to malignancy. However, further study is required to clarify HER-2 amplification using in situ hybridization. We also performed sequencing of PTEN and PIK3CA genes in tupaias No.75, 97, 185, and 211. However, we did not detect any mutations in these genes, as previously reported [12].

**A**

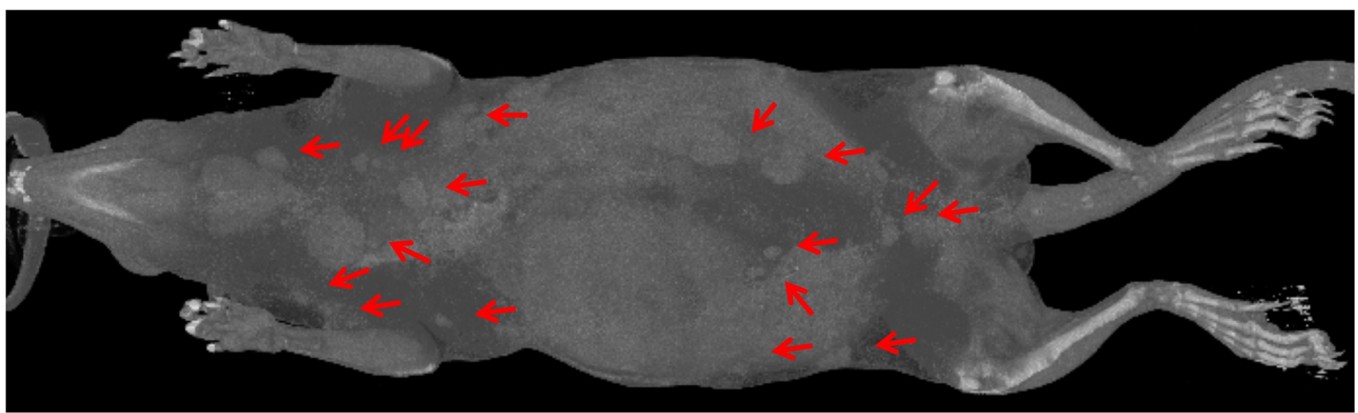

**B**                                                **C**

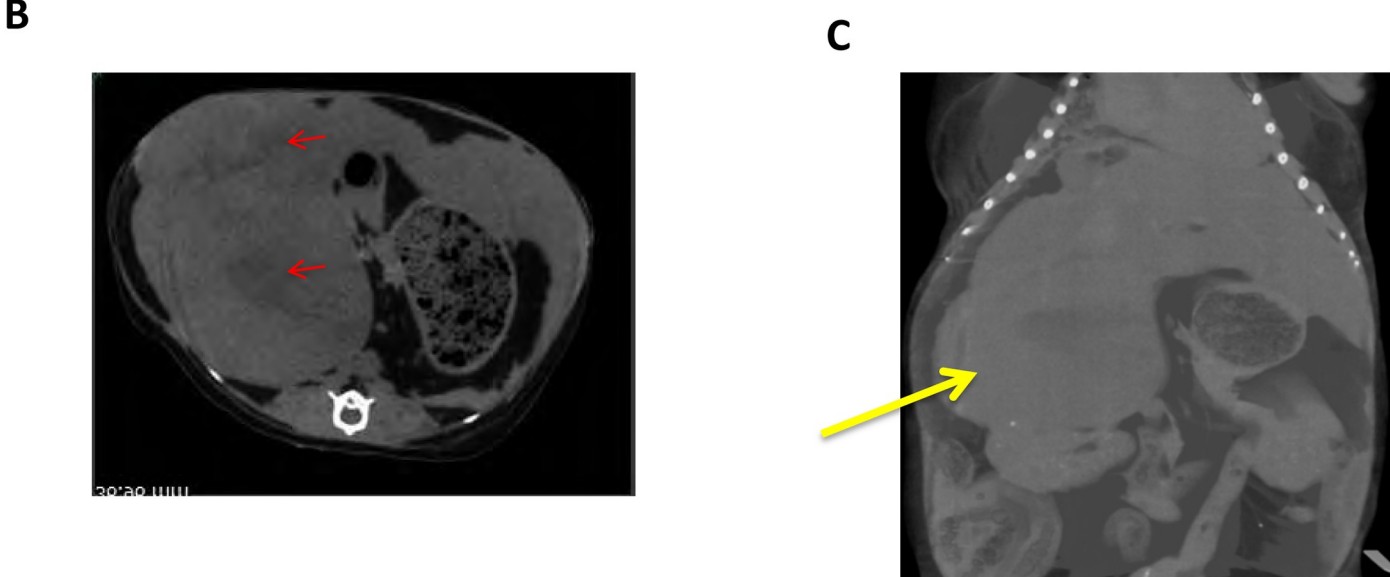

**Fig 7. X-ray CT (LCT-200) analysis of a tumor-bearing tree shrew (No.26, 4 years 11 months old).** (**A**) Three-dimensional (3D) analysis of whole tree shrew body. Position of sarcoma was indicated by red arrows. (**B**) 3D tomogram of tree shrew. Arrows indicating tumor tissues. (**C**) 3D analysis of tree shrew belly. Yellow arrows indicate the large intraperitoneal tumors.

Further study is required for clarification of the gene mutations in spontaneous tupaia mammary tumors, which could shed light on the genetical aspects for familial inheritance of mammary tumors.

In this study, simultaneous or metachronous multiplex tumors were frequently observed (9/15; 60%). Incidence of tumor was frequently observed in multiple mammary glands. Except one malignant case (No. 75), all tumors were noninvasive. Moreover, level of ER expression showed variations in multiple tumors in one individual (No. 97) (Table 2), suggesting that they did not share the site of origin and were not recurrent cases. Therefore, most of the simultaneous or metachronous multiplex tumors possibly originated from multiple sites.

Oncolytic virotherapy is one of the promising approaches for analyzing the therapeutic strategies of breast cancers [22, 27]. Tumor-expressing molecules like the epidermal growth

factor receptor (EGFR) can be a suitable target for cancer therapy [28–30]. Nectin-4 is a diagnostic and therapeutic target for lung cancer [31] and breast carcinoma [32], and the epithelial cell receptor of measles virus [33]. Based on these observations, the measles virus vector has been developed for various cancers [22, 27, 33]. We observed the expression of nectin-4 on the surface of tree shrew mammary tumor cells. Also, X-ray CT analysis has made possible the observation of tumor tissues in the body of tree shrews. Therefore, the tree shrew model should contribute towards the development of therapeutic strategy for mammary tumors in future studies.

## Supporting information

**S1 Data. Raw data of Fig 1 (onset of mammary tumors in female tree shrews) was indicated.**
(XLSX)

**S2 Data. Analysis on liver (2–1) and brain (2–2) of tree shrew #26 (4 years 11 months) by LCT-200.**
(PDF)

**S1 Fig.** (**A**) Outlook of CT LCT-200 (Hitachi Ltd., Tokyo Japan). (**B**) Whole body structure (upper) and bone structure (lower). (**C**) The skull structure of tree shrew (#26, 4 years 11 months).
(PDF)

**S2 Fig. H&E staining of intraperitoneal tumor of tree shrew #26.** (**A**) Tumor (upper, blue arrow) and liver (lower). (**B**) Tumor (blue arrow) and kidney (left). (**C**) Tumor (x40) (**D**) Tumor (x200), Bar-100μm.
(PDF)

**S1 Table. J-score.**
(DOCX)

**S2 Table. HER-2 score.**
(DOCX)

**S1 File.**
(DOCX)

## Acknowledgments

The authors thank Mr. Shinsuke Kimura (Hitachi Co.) for his help in performing the X-ray CT analysis of the tree shrew tissues. This work was supported by grants from the Japan Agency for Medical Research and Development (AMED), the Tokyo Metropolitan Government; the Ministry of Health and Welfare of Japan; and the Ministry of Education, Culture, Sports, Science and Technology of Japan.

## Author Contributions

**Data curation:** Chi Hai-Ying, Yuki Tanaka, Tatsuro Hifumi, Koichiro Shoji, Mohammad Enamul Hoque Kayesh, Tomoko Fujiyuki, Misako Yoneda, Akira Yabuki, Chieko Kai, Kyoko Tsukiyama-Kohara.

**Funding acquisition:** Michinori Kohara, Kyoko Tsukiyama-Kohara.

**Supervision:** Michinori Kohara, Kyoko Tsukiyama-Kohara.

**Writing – original draft:** Yuki Tanaka, Tatsuro Hifumi, Mohammad Enamul Hoque Kayesh, Md Abul Hashem, Bouchra Kitab, Takahiro Sanada, Hitoshi Hatai, Noriaki Miyoshi, Kyoko Tsukiyama-Kohara.

**Writing – review & editing:** Tatsuro Hifumi, Mohammad Enamul Hoque Kayesh, Michinori Kohara, Kyoko Tsukiyama-Kohara.

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
