## [Decision Letter · Decision Letter 0]

10 Mar 2020

PONE-D-20-04206

Pathological and genetic aspects of spontaneous mammary gland tumor in Tupaia belangeri (Tree shrew)

PLOS ONE

Dear Dr. Tsukiyama-Kohara,

Thank you for submitting your manuscript to PLOS ONE. After careful consideration, we feel that it has merit but does not fully meet PLOS ONE’s publication criteria as it currently stands. Therefore, we invite you to submit a revised version of the manuscript that addresses the points raised during the review process.  In particular, please ensure that pathological characterizations are complete and accurate and that driver gene alterations are established. 

We would appreciate receiving your revised manuscript by Apr 24 2020 11:59PM. To enhance the reproducibility of your results, we recommend that if applicable you deposit your laboratory protocols in protocols.io, where a protocol can be assigned its own identifier (DOI) such that it can be cited independently in the future. For instructions see: http://journals.plos.org/plosone/s/submission-guidelines#loc-laboratory-protocols

We look forward to receiving your revised manuscript.

Kind regards,

Yi Li, Ph.D.

Academic Editor

PLOS ONE

Journal Requirements:

2. We note that the tree shrews were not euthanised at the end of the study. In the Methods, please report the outcome of the animals; for example, if they were used in other studies, or were returned to the Laboratory Animal Centre.

Reviewers' comments:

Reviewer's Responses to Questions

**Comments to the Author**

1. Is the manuscript technically sound, and do the data support the conclusions?

Reviewer #1: Yes

Reviewer #2: Yes

Reviewer #3: Partly

2. Has the statistical analysis been performed appropriately and rigorously? 

Reviewer #1: N/A

Reviewer #2: Yes

Reviewer #3: N/A

3. Have the authors made all data underlying the findings in their manuscript fully available?

Reviewer #1: Yes

Reviewer #2: Yes

Reviewer #3: Yes

4. Is the manuscript presented in an intelligible fashion and written in standard English?

Reviewer #1: Yes

Reviewer #2: Yes

Reviewer #3: Yes

5. Review Comments to the Author

Reviewer #1: Authors bred and characterized 61 female tree shrews and identified 15 spontaneous mammary tumors. The incidence rate of mammary tumors was 24.6 % (15/61) and the average age of tumor incidence was 2 years 3 months. Unknown inherited factor(s) may paly a role in familial BC. Most cases were simple adenomas and two cases were tubulopapillary carcinomas. Over 90% cases were positive for both ERa and PR. These findings are quite interesting.

Major concerns:

1. Whether pTEN and PI3KCA genes are mutated in these spontaneous mammary tumors should be validated.

2. Ref 25 was incorrectly cited. Additionally, the following ref should be cited: characterization of spontaneous breast tumor in tree shrews (Tupaia belangeri chinenesis), Zoological Research, 2012，Feb. 33(1): 55−59.

3. The pathological differences between this study and Ref 12 should be discussed. Simple adenomas may not be accurate. Please consult pathologist for the classification.

Reviewer #2: Due to their structural differences between mouse and human mammary glands, commonly used mouse models of breast cancer have some limitations. Authors of this manuscript entitled “Pathological and genetic aspects of spontaneous mammary gland tumor in Tupaia belangeri (Tree shrew)” characterized the pathological and genetic aspects of spontaneous mammary tumors in Tupaia belangeri (tree shrew), which shows relatively high genetic homology and structural similarity to human mammary glands. Authors found that the tree shrews in their laboratory have high mammary tumor incidence rate (24.6%) and high rate of simultaneous or metachronous multiplex mammary tumors (60%). Some cases seemed to be of familial mammary gland tumors. Most tumors are benign adenoma and ER+. Two malignant mammary tumors (tubulopapillary carcinoma) were also observed. This work may provide a new animal model for breast cancer research. Future investigation on the genetic basis of familial cases may provide new insight into the mechanism of mammary tumorigenesis. The expression of Nectin-4, the receptor of measles virus, in some mammary tumor cells may provide a model to study the oncolytic virotherapy of breast cancer. The manuscript is overall well written. Following a few correction/modification may improve the manuscript.

1. Typo: “Nectin-4 has been also been reported” (page 11) should be corrected.

2. Figures 5 and 6 can be combined to save space.

3. Please provide a little bit more information in the Figure legend of Figure 1.

Reviewer #3: This interesting paper highlights pathological characteristics of another animal model of breast cancer, the tree shrew. While the pathological descriptions are evident, the manuscript does focus quite a bit on the familial inheritance of mammary tumors seen in their model. However the data supporting these observations is quite minimal aside from pedigree charting. Are these familial patterns driven by germline mutations that can be identified,particularly those associated with inherited forms of breast cancer. I think the genetic aspects should be explored more here for discussion, including sequencing of at least the major inherited drivers in breast cancer.

6. PLOS authors have the option to publish the peer review history of their article (what does this mean?). If published, this will include your full peer review and any attached files.

Reviewer #1: No

Reviewer #2: No

Reviewer #3: No

---

## [Author Response · Author response to Decision Letter 0]

27 Apr 2020

Reviewer #1: Authors bred and characterized 61 female tree shrews and identified 15 spontaneous mammary tumors. The incidence rate of mammary tumors was 24.6 % (15/61) and the average age of tumor incidence was 2 years 3 months. Unknown inherited factor(s) may paly a role in familial BC. Most cases were simple adenomas and two cases were tubulopapillary carcinomas. Over 90% cases were positive for both ERa and PR. These findings are quite interesting.

Major concerns:

1. Whether pTEN and PI3KCA genes are mutated in these spontaneous mammary tumors should be validated.

In line with the reviewer’s suggestions, we performed sequencing of PTEN and PIK3CA genes in the mammary tumors of tupaia No.75 (malignant, 6Y1M), No.97 (3Y11M), No.185 (2Y8M), and No.211 (10M), as reported previously (Xia et al., Eur. J. Cancer, 2014, reference #12). However, we did not observe any mutations in these genes, as reported in reference #12.

2. Ref 25 was incorrectly cited. Additionally, the following ref should be cited: characterization of spontaneous breast tumor in tree shrews (Tupaia belangeri chinenesis), Zoological Research, 2012，Feb. 33(1): 55−59.

In line with the reviewer’s comments, we have corrected reference 25 by replacing it with reference 12, and included the reference suggested by the reviewer [Characterization of spontaneous breast tumor in tree shrews (Tupaia belangeri chinenesis), Zoological Research, 2012，Feb. 33(1): 55−59].

3. The pathological differences between this study and Ref 12 should be discussed. Simple adenomas may not be accurate. Please consult pathologist for the classification.

From the results of this study, mutations in PTEN and PIK3CA genes may not always be induced in spontaneous mammary tumors in tupaia (p11, lines 12-13). However, further study is required in this field. We have discussed these points on p14, lines 7-10. 

We have discussed the pathological differences with a veterinary pathologist and changed “simple adenomas” to “intraductal papillary adenoma (duct papilloma)(IPA)”, and included reference 15. We have also explained these findings in greater detail (p10, lines 11–15).

Reviewer #2: Due to their structural differences between mouse and human mammary glands, commonly used mouse models of breast cancer have some limitations. Authors of this manuscript entitled “Pathological and genetic aspects of spontaneous mammary gland tumor in Tupaia belangeri (Tree shrew)” characterized the pathological and genetic aspects of spontaneous mammary tumors in Tupaia belangeri (tree shrew), which shows relatively high genetic homology and structural similarity to human mammary glands. Authors found that the tree shrews in their laboratory have high mammary tumor incidence rate (24.6%) and high rate of simultaneous or metachronous multiplex mammary tumors (60%). Some cases seemed to be of familial mammary gland tumors. Most tumors are benign adenoma and ER+. Two malignant mammary tumors (tubulopapillary carcinoma) were also observed. This work may provide a new animal model for breast cancer research. Future investigation on the genetic basis of familial cases may provide new insight into the mechanism of mammary tumorigenesis. The expression of Nectin-4, the receptor of measles virus, in some mammary tumor cells may provide a model to study the oncolytic virotherapy of breast cancer. The manuscript is overall well written. Following a few correction/modification may improve the manuscript.

1. Typo: “Nectin-4 has been also been reported” (page 11) should be corrected.

We have corrected this sentence (p11, line 5 from the bottom).

2. Figures 5 and 6 can be combined to save space.

We have combined Figures 5 and 6.

3. Please provide a little bit more information in the Figure legend of Figure 1.

In line with the reviewer’s comments, we have updated the figure legend with more information.

Reviewer #3: This interesting paper highlights pathological characteristics of another animal model of breast cancer, the tree shrew. While the pathological descriptions are evident, the manuscript does focus quite a bit on the familial inheritance of mammary tumors seen in their model. However the data supporting these observations is quite minimal aside from pedigree charting. Are these familial patterns driven by germline mutations that can be identified,particularly those associated with inherited forms of breast cancer. I think the genetic aspects should be explored more here for discussion, including sequencing of at least the major inherited drivers in breast cancer.

We thank the reviewer for their helpful comments. In line with these comments, we sequenced the PTEN and PIK3CA genes in different tupaia tumors [tupaia No.75 (malignant, 6Y1M), No.97 (3Y11M), No.185 (2Y8M), and No.211 (10M)]. However, we did not detect any mutations in these genes, which was reported previously (reference 12). We have discussed these findings in the discussion (p14, lines 7–11).

---

## [Editor Report · Decision Letter 1]

1 May 2020

Pathological and genetic aspects of spontaneous mammary gland tumor in Tupaia belangeri (Tree shrew)

PONE-D-20-04206R1

Dear Dr. Tsukiyama-Kohara,

We are pleased to inform you that your manuscript has been judged scientifically suitable for publication and will be formally accepted for publication once it complies with all outstanding technical requirements.

With kind regards,

Yi Li, Ph.D.

Academic Editor

PLOS ONE
---

## [Editor Report · Acceptance letter]

7 May 2020

PONE-D-20-04206R1 

Pathological and genetic aspects of spontaneous mammary gland tumor in Tupaia belangeri (Tree shrew) 

Dear Dr. Tsukiyama-Kohara:

I am pleased to inform you that your manuscript has been deemed suitable for publication in PLOS ONE. Congratulations! Your manuscript is now with our production department. 

With kind regards,

on behalf of

Dr. Yi Li 

Academic Editor

PLOS ONE